# X-linked neonatal-onset epileptic encephalopathy associated with a gain-of-function variant p.R660T in *GRIA3*

Jia-Hui Sun[1,2☯], Jiang Chen[1☯], Fernando Eduardo Ayala Valenzuela[3☯], Carolyn Brown[4], Diane Masser-Frye[5], Marilyn Jones[5], Leslie Patron Romero[6], Berardo Rinaldi[7], Wenhui Laura Li[8], Qing-Qing Li[1,2], Dan Wu[1,2], Benedicte Gerard[9], Erin Thorpe[4,12]*, Allan Bayat[10,11]*, Yun Stone Shi[1,2,12,13]*

**1** Ministry of Education Key Laboratory of Model Animal for Disease Study, Department of Neurology, Affiliated Drum Tower Hospital, Medical School, Nanjing University, Nanjing, China, **2** State Key Laboratory of Pharmaceutical Biotechnology, Model Animal Research Center, Nanjing University, Nanjing, China, **3** Hospital Angeles Tijuana, Tijuana, México, **4** Illumina Inc., San Diego, California, United States of America, **5** Division of Genetics, Department of Pediatrics, UC San Diego School of Medicine, Rady Children's Hospital, San Diego, California, United States of America, **6** Facultad de Medicina y Psicología, Universidad Autónoma de Baja California, Tijuana, Mexico, **7** Fondazione IRCCS Ca' Granda Ospedale Maggiore Policlinico, Milan, Italy, **8** Breakthrough Genomics Inc., Irvine, California, United States of America, **9** Laboratoires de diagnostic génétique, Institut Medical d'Alsace, Hôpitaux Universitaire de Strasbourg, Strasbourg, France, **10** Department for Genetics and Personalized Medicine, Danish Epilepsy Centre, Dianalund, Denmark, **11** Institute for Regional Health Services Research, University of Southern Denmark, Odense, Denmark, **12** Institute for Brain Sciences, Nanjing University, Nanjing, China, **13** Chemistry and Biomedicine Innovation Center, Nanjing University, Nanjing, China

☯ These authors contributed equally to this work.
* ethorpe@illumina.com (ET); abaya@filadelfia.dk (AB); yunshi@nju.edu.cn (YSS)

**Editor:** wei zhang, Hebei Medical University, CHINA

**Data Availability Statement:** All relevant data are within the manuscript and its Supporting Information files.

**Funding:** This work is supported by grants from National Key R & D Program of China

## Abstract

The X-linked *GRIA3* gene encodes the GLUA3 subunit of AMPA-type glutamate receptors. Pathogenic variants in this gene were previously reported in neurodevelopmental diseases, mostly in male patients but rarely in females. Here we report a *de novo* pathogenic missense variant in *GRIA3* (c.1979G>C; p. R660T) identified in a 1-year-old female patient with severe epilepsy and global developmental delay. When exogenously expressed in human embryonic kidney (HEK) cells, GLUA3_R660T showed slower desensitization and deactivation kinetics compared to wildtype (wt) GLUA3 receptors. Substantial non-desensitized currents were observed with the mutant but not for wt GLUA3 with prolonged exposure to glutamate. When co-expressed with GLUA2, the decay kinetics were similarly slowed in GLUA2/A3_R660T with non-desensitized steady state currents. In cultured cerebellar granule neurons, miniature excitatory postsynaptic currents (mEPSCs) were significantly slower in R660T transfected cells than those expressing wt GLUA3. When overexpressed in hippocampal CA1 neurons by *in utero* electroporation, the evoked EPSCs and mEPSCs were slower in neurons expressing R660T mutant compared to those expressing wt GLUA3. Therefore our study provides functional evidence that a gain of function (GoF) variant in *GRIA3* may cause epileptic encephalopathy and global developmental delay in a female subject by enhancing synaptic transmission.

(2019YFA0801603 to Y.S.S.), the National Natural Science Foundation of China (91849112 and 31571060 to Y.S.S., and 81901161 to C.J.), the Natural Science Foundation of Jiangsu Province (BE2019707 to Y.S.S.) and Fundamental Research Funds for the Central Universities (0903-14380029 to Y.S.S.). The funders had no role in study design, data collection and analysis, decision to publish, or preparation of the manuscript.

**Competing interests:** The authors have declared that no competing interests exist.

## Author summary

Glutamate is the excitatory neurotransmitter in brain, abnormality of which causes excito-toxicity and diseases. Here we identified a pathogenic missense variant in *GRIA3* gene in a female patient with severe epilepsy and global developmental delay. The X-linked *GRIA3* gene encodes GLUA3, a subunit of glutamate receptors. Through electrophysiological analysis of the mutant GLUA3 in a cell line and mouse neurons, we found this mutant makes strengthened glutamate receptors. This study thus indicates that the variant causes epileptic encephalopathy and global developmental delay by enhancing glutamate signaling in brain.

## Introduction

Alpha-amino-3-hydroxy-5-methyl-4-isoxazole propionate glutamate receptors (AMPARs) mediate the majority of fast excitatory synaptic transmission in brain and are encoded by the four known *GRIA* genes. Pathogenic variants in *GRIA1*, *GRIA2*, *GRIA3*, and *GRIA4* have been reported in individuals with various neurodevelopmental disorders, mostly intellectual disability (ID) and autistic features [1–4]. The X-linked *GRIA3* gene encodes GLUA3, one of the four pore-forming AMPAR subunits. Though GLUA3 alone can form $Ca^{2+}$ permeable homomeric receptors, in the brain it mainly forms heteromeric receptors with GLUA2 [5]. The GLUA2/A3 receptors contain an arginine residue at the channel pore of GLUA2 subunits due to a post-transcriptional editing [6–8], blocking $Ca^{2+}$ permeability [5,9–11].

Thus far, about 20 pathogenic variants in *GRIA3* have been reported, as summarized in a recent paper [12]. The majority of affected individuals are males with the pathogenic variant inherited from unaffected mothers [12,13]. However, pathogenic alterations in *GRIA3* have been reported in two female individuals as well. An early work found that a female with bipolar disorder and intellectual disability carried a balanced translocation affecting *GRIA3* [14]. Recently, a young female with early-onset epileptic encephalopathy was reported to harbor a *de novo* variant, p.A248V [15]. However, these two papers presented no functional testing that could support the clinical findings. Therefore functional analyses are needed to explore if and why single nucleotide variants in *GRIA3* cause disease in females.

In this study, a *de novo* variant in *GRIA3* was identified by clinical whole genome sequencing in a 1-year-old female with severe epilepsy and global developmental delay. We then systemically studied the physiological function of this mutant in HEK cells and neurons to determine the potential impact of this variant on protein function. The mutant GLUA3_R660T displayed slower desensitization and deactivation kinetics in HEK cells and neurons, suggesting it is a Gain-of-Function variant. Our data demonstrates the R660T mutant causes prolonged activity of AMPARs in brain, explaining the potential mechanisms in epileptogenesis.

## Results

### Clinical findings in proband with GLUA3_R660T

A currently 4-year-old girl was born at term to unrelated parents of Mexican heritage. The pregnancy was complicated by gestational diabetes and by early warfarin exposure due to mother's history of an atrial septal defect with stroke. Birth weight was 2.67 kg (-1.5 SD) and birth length was 47 cm (-1.5 SD). She had normal Apgar scores but experienced epileptic seizures within a few hours of birth. She was also hypertonic between seizures and only relaxed

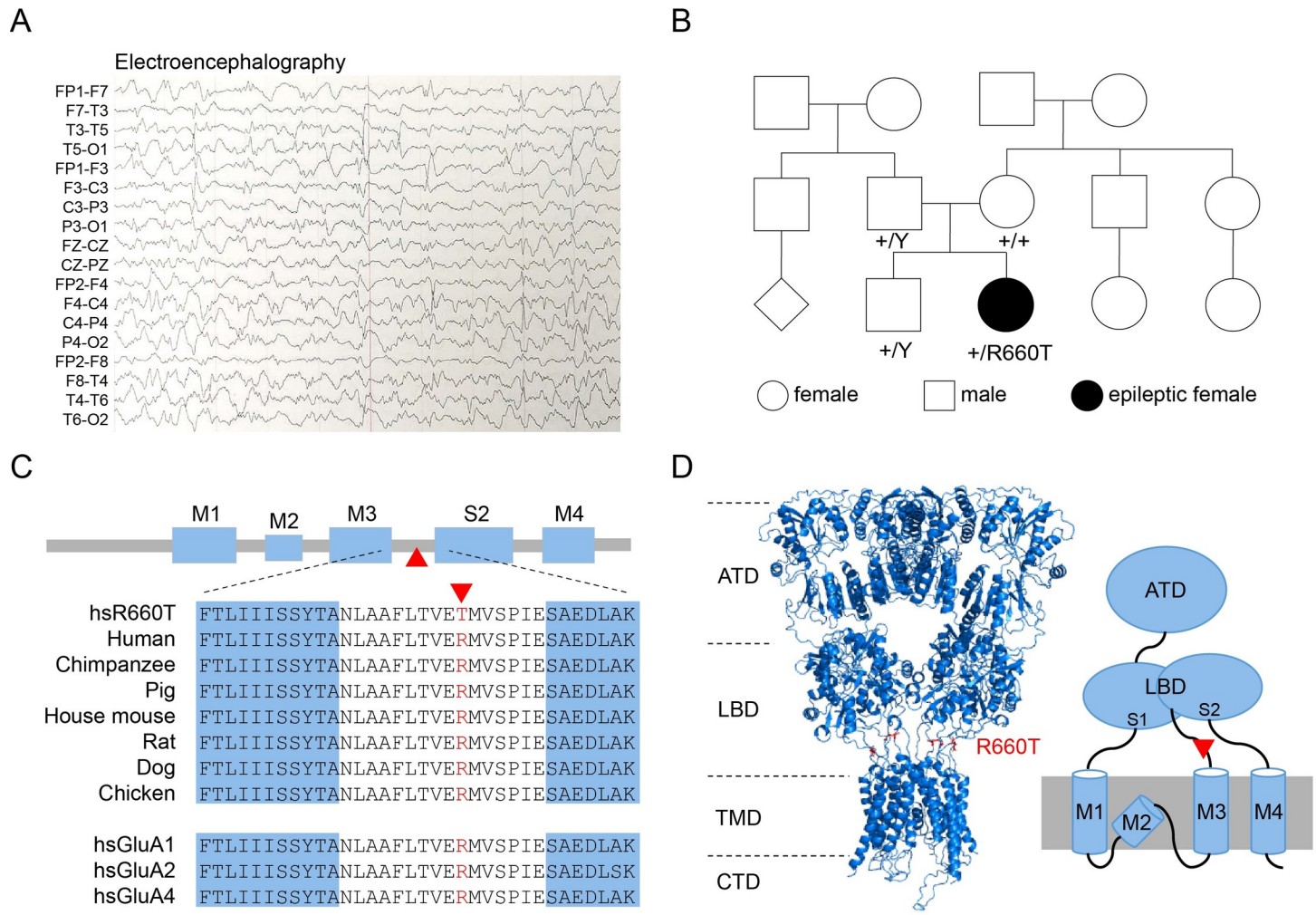

**Fig 1. The patient and the variant in *GRIA3*.** (A) Electroencephalography of the proband showing typical epileptic waveforms. (B) Family pedigree showing a *de novo GRIA3* variant (c.1979G>C, p.R660T) identified in the proband. The variant is absent in her parents and brother. (C) AMPA receptors architecture and sequence alignment. (D) A tertiary GLUA3 homomeric model (left) constructed from the GluA2 structure (PDB ID: 5WEO) and a schematic depict of a single GluA3 subunit (right) indicate the location of the mutant host residues.

when asleep. She was intubated for 20 days and remained in the neonatal intensive care unit for 30 days. An electroencephalography confirmed a diagnosis of epilepsy (Fig 1A) but due to limited access to her medical files, we were unable to obtain further clinical data regarding the neonatal period. She did not pass the newborn hearing screen but at 13 months her follow-up brainstem audiogram was normal.

At five months of life, she was treatment-resistant despite being on three anticonvulsants (vigabatrin, phenobarbital and valproate) and still had many daily brief myoclonic jerks along with several weekly bilateral tonic clonic seizures. She remained hypertonic and had brisk deep tendon reflexes. She did not show any facial dysmorphism but had a large inguinal hernia. The hernia recurred after initial repair. An ophthalmological examination showed cortical vision impairment and nystagmus. A brain magnetic resonance imaging performed in the newborn period was of poor quality but showed some thinning of the corpus callosum.

She is kept on levetiracetam, valproate acid, vigabatrin, cannabis oil, and clobazam but remains treatment-resistant. She still has both myoclonic jerks and bilateral tonic-clinic

seizures. Nevertheless, epilepsy burden has reduced from 30 to 10 seizures per day and convulsions are also shorter in duration. She is unable to feed orally and depends on a gastrostomy-tube. In addition, she presents with profound developmental delay. Currently, she has some head control but cannot support herself in a seated position and has severe growth restriction.

Whole genome sequencing revealed a heterozygous *de novo* missense variant in the *GRIA3* gene (NM_000828.4:c.1979G>C; p.R660T) (Fig 1B) when the patient was 1-year old. This variant affects a highly conserved amino acid (Fig 1C) and is located in the linker between the third transmembrane domain (M3) and the S2 extracellular domain of glutamate binding domain of AMPAR subunit (Fig 1C and 1D) [16]. Bioinformatic predictions are in favor of its pathogenicity (SIFT pathogenic (score 0); polyphen2, probably damaging (score 0,985). This variant was absent from the Genome Aggregation Database (gnomAD) and was not present the ClinVar or HGMDpro databases at the time of analysis. No additional variants of clinical interest in genes associated with epilepsy or other aspects of the patient's phenotype were identified.

## GLUA3_R660T mutant slowed deactivation and desensitization kinetics

To evaluate the effects of the GLUA3_R660T on functional properties of AMPA receptors, we transfected HEK293 cells with expression constructs of human GLUA3 cDNA or R660T variant. To visit the transfection, EGFP was linked to GLUA3 by an Internal Ribosome Entry Site (GLUA3-IRES-EGFP) to express GLUA3 and EGFP separately. Outside-out patched were excised and saturating concentration of glutamate (10 mM) was applied by a fast piezoelectric system [17]. It is known that glutamate release into synaptic cleft is with the concentration of millimolar and reabsorbed with time frame about 1 ms under normal physiological condition [18]. Therefore, the deactivation kinetics recorded by a 1 ms brief application of glutamate mimics the synaptic released glutamate in brain. The deactivation kinetics was slower for GluA3_R660T than wt GLUA3 (Fig 2A). When the extracellular glutamate clearance is impaired under pathophysiological conditions such as in epilepsy, the postsynaptic glutamate receptors are exposed in high concentration of glutamate for prolonged period of time. The desensitization kinetics recorded by prolonged exposure (500 ms) to glutamate would mimic this condition. The desensitization kinetics of R660T was also slower than that of wt GLUA3 (Fig 2B). Furthermore, R660T produced a substantial non-desensitized steady-state current, which was about 30% of the peak current and absent in wt GLUA3 (Fig 2B).

We further wondered whether the R660T had an effect on single channel conductance. Non-stationary fluctuation analysis (NSFA) provides a convenient method to measure single channel conductance [19]. We found that neither single channel conductance nor the peak open probability was affected by R660T (Fig 2C–2E).

## GLUA3_R660T mutant slows gating kinetics of GLUA2/A3 receptor

In the brain, GLUA3 forms heteromeric Ca$^{2+}$-impermeable AMPARs with GLUA2 [5]. The current-voltage (I-V) curves of GLUA3 homomers are rectifying and GLUA2/A3 heteromers are linear [20–22]. To examine whether the R660T variant affects the capability of GLUA3 to form heteromeric receptors, the mutant was co-expressed with edited GLUA2 subunits (GLUA2R) in HEK cells. In the absence of GLUA2, the rectification of the mutant was the same as wt GLUA3 (Fig 3A). In the presence of GLUA2, wt GLUA3 showed a linear IV-curve, indicating GLUA2/A3 are the major receptors under the condition. GLUA3_R660T co-expression with GLUA2 also showed a linear IV-curve with identical rectification index as the wt GLUA2/A3, demonstrating that the mutant did not change its capability to form heteromeric receptors with GLUA2 (Fig 3B). We then examined the gating kinetics of GLUA2/A3

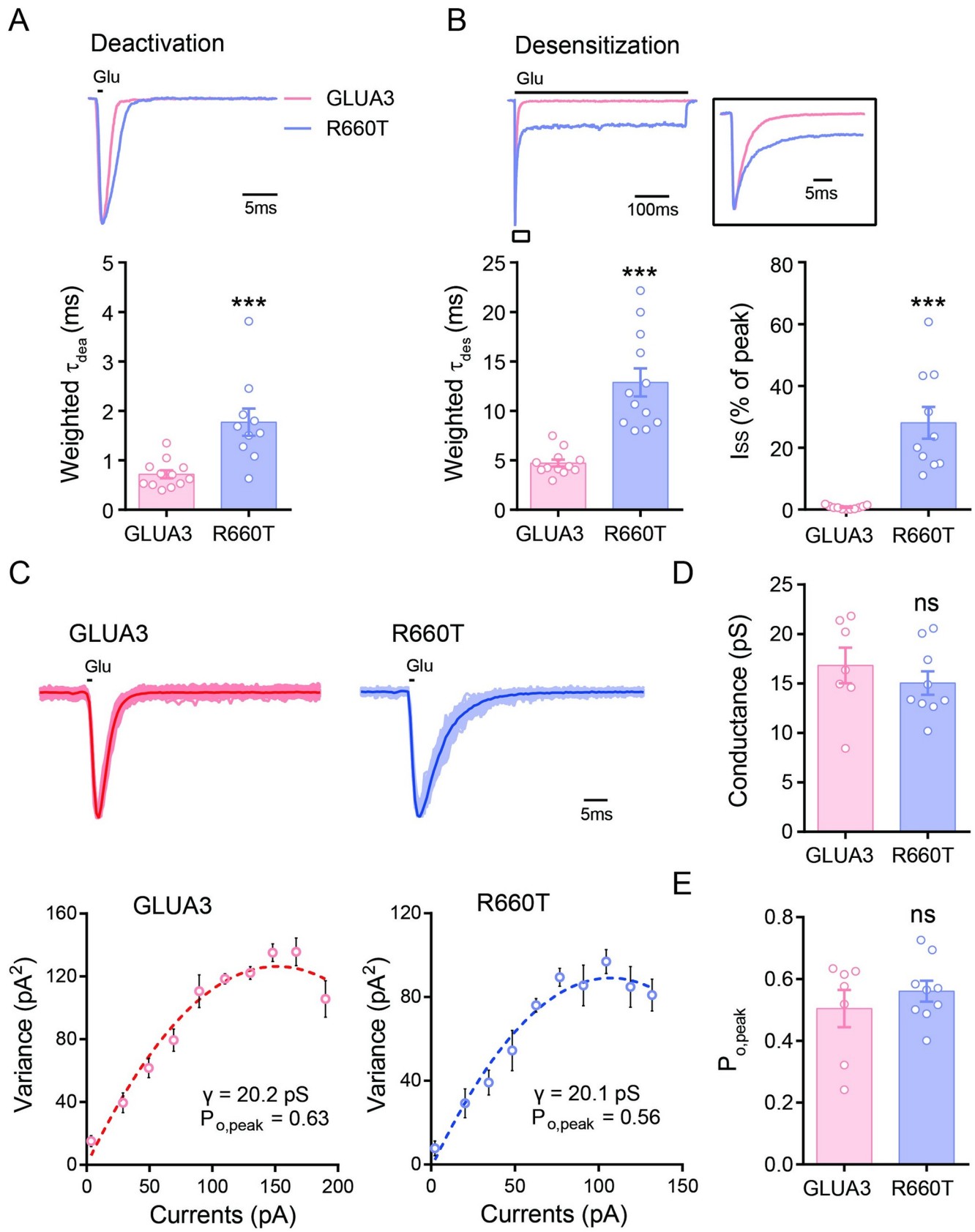

**Fig 2. R660T variant slows deactivation and desensitization of GLUA3.** (A) Deactivation of GLUA3 and GLUA3_R660T expressed in HEK cells. Up panel, glutamate (10 mM) applied to outside-out patches excised from transfected HEK cells. Low panel, bar graph showed the deactivation slowed by the R660T variant. GLUA3, $0.7 \pm 0.1$ ms, n = 12; R660T, $1.8 \pm 0.3$ ms, n = 10; ***p < 0.001, unpaired t-test. (B) Desensitization of GLUA3 and R660T variant. Up panel, desensitization curves. Boxed, close up of the desensitization curves around the peak region. Low left, the statistics of weighted $\tau_{des}$. GLUA3, $4.7 \pm 0.4$ ms, n = 12; R660T, $12.9 \pm 1.4$ ms, n = 12; ***p < 0.001. Low right, the steady state currents normalized peak value. GLUA3, $0.7 \pm 0.2\%$, n = 11; R660T, $28.1 \pm 5.2\%$, n = 10; ***p < 0.001. (C) Non-steady fluctuation analysis. Up, the row traces of multiple recordings from same patches. Highlighted shows average traces. Low panel, non-steady fluctuation analysis for above recordings. (D) The statistics of single channels conductance calculated from non-steady fluctuation analysis. GLUA3, $16.8 \pm 1.8$ pS, n = 7; R660T, $15.0 \pm 1.2$ pS, n = 9; ns, p = 0.405. (E) The peak open probability. GLUA3, $0.50 \pm 0.06$, n = 7; R660T, $0.56 \pm 0.03$, n = 9; ns, p = 0.401. Data are presented as mean ± SEM. Unpaired t-test was used for data analysis.

heteromers, and found that the deactivation and desensitization kinetics of GLUA2/A3 receptors were also slowed by the R660T mutant (Fig 3C and 3D). In addition, GLUA2/A3_R660T displayed non-desensitized steady state currents during prolonged exposure to glutamate (Fig 3D). In summary, these data suggested that the mutant primarily slowed AMPAR gating kinetics.

## The effects of auxiliary subunits on the mutant

The native AMPA receptors are associated with auxiliary subunits in the brain, including transmembrane AMPA receptor regulatory proteins (TARPs), cornichon-like (CNIH), germ cell-specific gene 1-like (Gsg1l) and the cysteine-knot AMPAR-modulating protein (CKAMP)/Shisa protein family [23–26]. These auxiliary subunits regulate gating kinetics and synaptic function of AMPARs [27–29]. We then coexpressed GLUA3 with the prototypical TARP γ-2, the deactivation and desensitization kinetics were slowed, as expected, by TARP γ-2 (Fig 4A and 4B). The deactivation and desensitization of R660T was also slowed by TARP γ-2 (Fig 4A and 4B). The non-desensitized currents of GLUA3_R660T/γ-2 was increased compared to GLUA3/γ-2 (Fig 4B). We further measured the gating kinetics of heteromeric GLUA2/A3 in the presence of TARP γ-2. Similarly, deactivation and desensitization were slowed by TARP γ-2, while R660T further slowed these kinetics (Fig 4C and 4D). The non-desensitized currents of GLUA2/A3_R660T/γ-2 was increased compared to GLUA2/A3/γ-2 (Fig 4D).

We also examined cornichon family AMPA receptor auxiliary protein 2 (CNIH2), another auxiliary subunit of AMPARs, on the gating kinetics on GLUA3_R660T. We found that the deactivation and desensitization of GLUR3 with or without GLUA2 were slowed by CNIH2 (S1 Fig), while R660T further enhanced the slowing.

## GLUA3_R660T variant slows the mini-EPSCs in cerebellar granule cells

The slower kinetics of the mutant GLUA3 suggests it should affect synaptic AMPAR function. Primary culture of cerebellar granule neurons (CGNs) is a convenient system to record miniature AMPAR-EPSCs [30,31]. We therefore isolated and cultured primary CGNs from P6-8 mice. Two days after division (DIV 2), the neurons were transfected with wt or mutant GLUA3. At DIV 9–10, miniature EPSCs (mEPSCs) were recorded from the transfected CGNs or the naïve control neurons. We found that the mEPSCs in neurons transfected with wt GLUA3 decayed faster than untransfected neurons (Fig 5A and 5B), indicating that the transfected GLUA3 had successfully targeted to synapses. Furthermore, the data indicate that overexpressed GLUA3 are faster than the endogenous AMPA receptors. In the contrast, mEPSCs in GLUA3_R660T expressing CGNs were slower than untransfected neurons and wt GLUA3 transfected neurons (Fig 5A and 5B). These data demonstrate that GLUA3_R660T slows the kinetics of synaptic AMPARs.

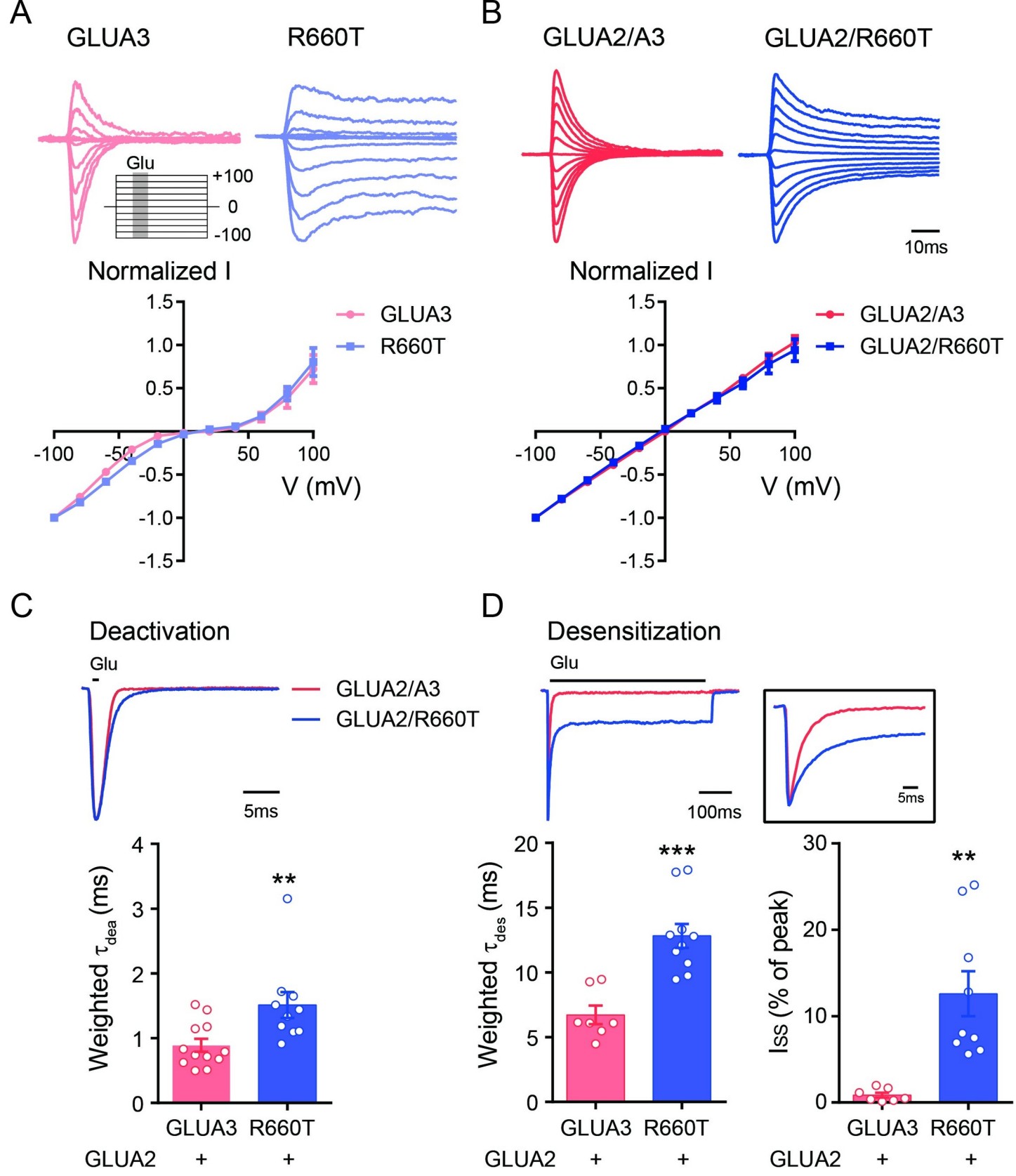

**Fig 3. GLUA3_R660T variant slowed the deactivation and desensitization of heteromeric GluA2/A3 receptors.** (A) I-V relationship for GLUA3 and R660T homomeric receptors. Up panel, Desensitization curves were recorded while the holding potential was elevated from -100 mV with a step of 20 mV to +100 mV. Insert shows the voltage protocol. Dark area represents drug application (10 mM glutamate for 200 ms). Low panel, I-V curve. Peak currents were normalized to the absolute value of the peak current amplitudes recorded at -100 mV. (B) I-V relationship for GLUA2/A3 and GLUA2/A3_R660T. (C) The deactivation of GLUA2/A3 receptors was slowed by R660T variant. GLUA2/A3, $0.9 \pm 0.1$ ms, n = 12; GLUA2/A3_R660T, $1.5 \pm 0.2$ ms, n = 11; [**]p = 0.0083. (D) Up, the sample traces for desensitization of GLUA2/A3 and GLUA2/A3_R660T. Low left, statistics of weighted $\tau_{des}$. GLUA2/A3, $6.7 \pm 0.7$ ms, n = 7; GLUA2/A3_R660T, $12.8 \pm 0.9$ ms, n = 10; [***]p < 0.001. Low right, statistics of steady state currents. GLUA2/A3, $0.9 \pm 0.3\%$, n = 7; GLUA2/A3_R660T, $12.6 \pm 2.6\%$, n = 9; [**]p = 0.0015. Data are presented as mean ± SEM. Unpaired t-test was used for data analysis.

## GLUA3_R660T variant slows evoked AMPAR-EPSCs in hippocampal neurons

We then wondered if the GLUA3 variant affects the AMPAR function *in vivo*. To test this possibility, we transfected wt or mutant GLUA3 into hippocampal neurons via *in utero* electroporation. Acute hippocampal slices were prepared from P21-P28 puppies. Synaptic functions of glutamate receptors were analyzed by dual whole-cell recordings on a transfected neuron and a neighboring control neuron simultaneously (Fig 6A). We found that overexpression of wt GLUA3 reduced the peak amplitudes of evoked AMPAR-EPSCs (AMPAR-eEPSCs) while the GLUA3_R660T did not (Fig 6B–6E). However, the AMPAR-mediated charge transfer was increased by R660T compared to the control neurons (Fig 6F), indicating that the receptor channel may open longer. Indeed, analysis of the decay kinetics of AMPAR-eEPSCs showed that wt GLUA3 speeded up the decay of AMPAR-eEPSCs while the R660T slowed it (Fig 6G). Consistently, the mEPSCs were speeded up by wt GLUA3 while slowed by R660T (Fig 6H). Paired pulse ratio (PPR), a measure of the release probability of presynaptic neurotransmitters in neurons, was unaltered by overexpression of GLUA3 and R660T (Fig 6I). NMDAR-eEPSCs were unaltered in GLUA3 and R660T transfected neurons (Fig 6J–6M). These data thus provide evidence supporting that the R660T variant slows the decay of synaptic AMPARs *in vivo*.

## Discussion

Since the first description, about 20 *GLUA3* pathogenic variants have been reported, including a balanced translocation in which the breakpoints disrupted the *GRIA3* gene, a deletion, duplications, and missense variants. Most of them are found in males with X-linked ID together possibly associated with dysmorphic features or epilepsy, and are inherited from unaffected mothers [12,13]. Two *de novo* variants have been reported in females, one with bipolar symptom and ID who carries a balanced translocation involving *GRIA3* [14], and another with epilepsy for whom a *de novo* p.A248V variant was identified [15]. Our case is the third affected female to be reported. She presented with a devastating neurological phenotype compatible with a developmental and epileptic encephalopathy. Symptoms included treatment resistant neonatal-onset epilepsy, congenital hypertonia, and severe developmental delay.

The variant affects a highly conserved amino acid (R660T) and is located in the extracellular linker 2 of AMPAR, between M3 transmembrane domain and S2 glutamate binding domain [16]. Recombinant expression in HEK cells showed that the R660T variant causes slowing of deactivation and desensitization of GLUA3 homomeric receptors as well as GLUA2/A3 heteromeric receptors. By co-expression with TARP γ-2 and CNIH2, we also demonstrate that the slowing of channel kinetics are further enhanced by AMPAR auxiliary subunits TARPs and cornichons [19,32]. When overexpressed in both cultured CGNs and hippocampal neurons, the variant slows the decay kinetics of miniature and evoked AMPAR-EPSCs, consistent with the observations in HEK cells. Interestingly, an early study demonstrates that mutations on the corresponding sites in rodent GluA1, GluA2 and kainate-type glutamate receptor GluK2 slow the ion channel kinetics [33]. Our study on GLUA3_R660T thus provides further

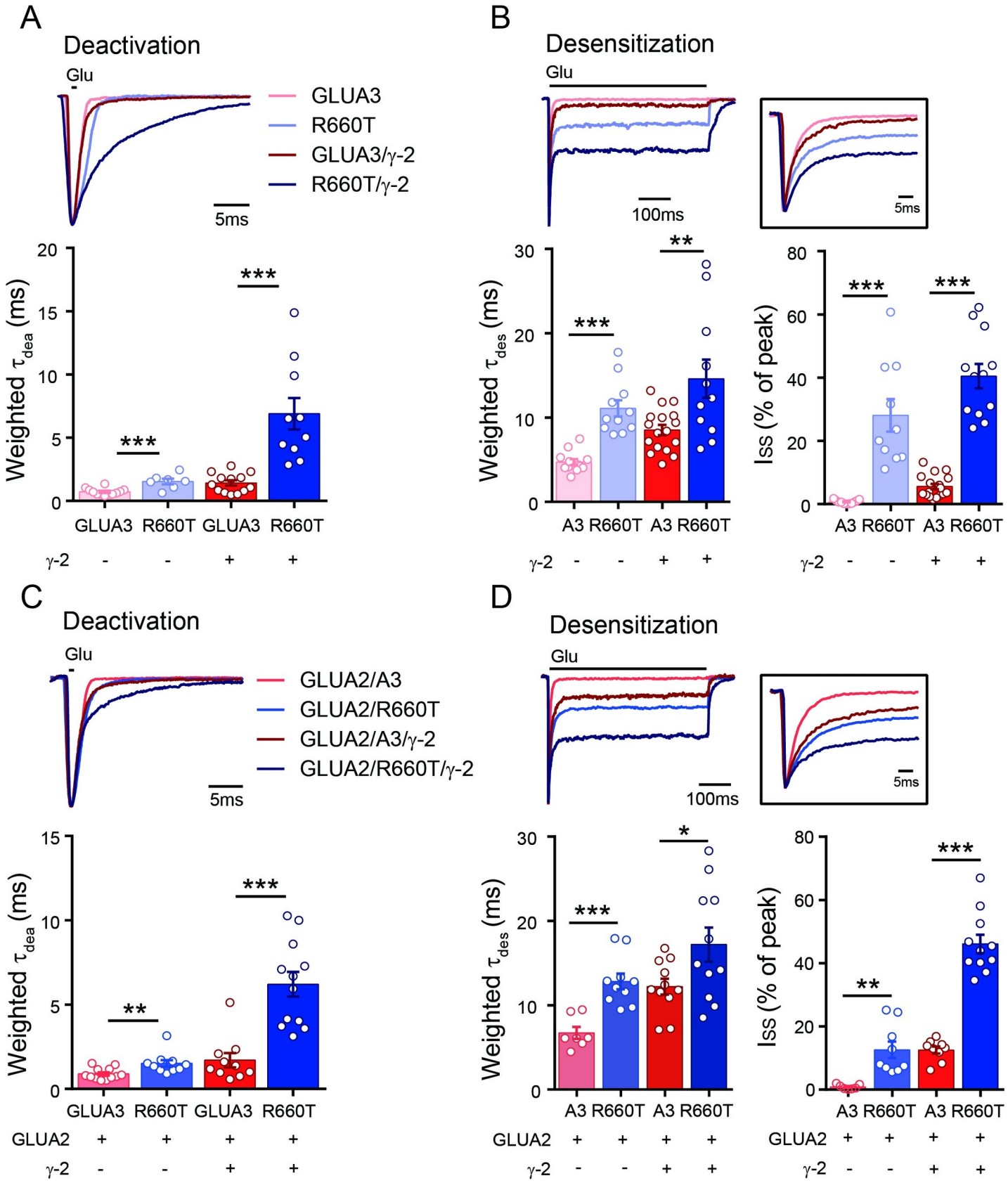

**Fig 4. The deactivation and desensitization kinetics of GLUA3 and GLUA2/A3 and variant in the presence of TARP γ-2.** (A). Deactivation of GLUA3 and GLUA3_R660T in the presence of γ-2. GLUA3 and GLUA3_R660T were the same as in Fig 1. GLUA3/γ-2, 1.4 ± 0.2 ms, n = 13; R660T/γ-2, 6.9 ± 1.3 ms, n = 10; ***p < 0.001. (B) Up, the sample traces for desensitization of GLUA2/A3 and GLUA2/A3_R660T in the presence of γ-2. Low left, statistics of weighted $\tau_{des}$. GLUA3/γ-2, 8.5 ± 0.6 ms, n = 17; R660T/γ-2, 14.6 ± 2.3 ms, n = 11; ** p = 0.0047. Low right, statistics of steady state currents. GLUA3/γ-2, 5.6 ± 0.9%, n = 17; R660T/γ-2, 40.5 ± 3.8%, n = 12; ***p < 0.001. (C) Deactivation of GLUA2/A3 and GLUA2/A3_R660T in the presence of γ-2. GLUA2/A3 and GLUA2/A3_R660T were the same as in Fig 2. GLUA2/A3/γ-2, 1.7 ± 0.4 ms, n = 10; GLUA2/A3_R660T/γ-2, 6.2 ± 0.7 ms, n = 12; ***p<0.001. (D) Up, the sample traces for desensitization of GLUA2/A3/γ-2 and GLUA2/A3_R660T/γ-2. Low left, statistics of weighted $\tau_{des}$. GLUA2/A3/γ-2, 12.2 ± 0.9 ms, n = 11; GLUA2/A3_R660T/γ-2, 17.2 ± 2.0 ms, n = 12; *p = 0.0374. Low right, statistics of steady state currents. GLUA2/A3/γ-2, 12.5 ± 1.1%, n = 9; GLUA2/A3_R660T/γ-2, 46.1 ± 2.9%, n = 11; ***p < 0.001. Data are presented as mean ± SEM. Unpaired t-test was used for data analysis.

evidence that the linker region between M3 and S2 domains plays an important role in AMPAR gating. Most importantly, our experiments demonstrate that the p.(R660T) variant shows a GoF effect on AMPARs.

It is widely accepted that glutamate-medicated hyperexcitability of neural circuits plays a causative role in seizure generation [34,35]. Intracerebral injection of glutamate or glutamate receptor agonists into laboratory animals causes epileptic seizures [36]. Disturbance of extra-cellular glutamate clearance also causes epilepsy [37]. Cyclothiazide, a potent AMPAR desensitization blocker, induces seizure in rodents [38], suggesting that enhancing postsynaptic glutamate receptor function also leads to epilepsy. Recently variants in NMDA receptors and AMPARs have been reported to cause epilepsy [39,40]. Such AMPAR dynamic anomalies could be secondarily reinforced and worsened as epileptic seizures cause fast release and extra-cellular accumulation of glutamate, which further induces excitotoxicity and neural damage [9,35].

Treatment of epilepsy remains largely empirical, and individual prescribing based on the mechanism of action is generally not possible. However, recent findings in genetic epilepsies have elucidated some mechanisms of epileptogenesis, unravelling the role of a number of genes with different functions, such as ion channels, proteins associated to the vesicle synaptic cycle or involved in energy metabolism. The advent of Next Generation Sequencing is

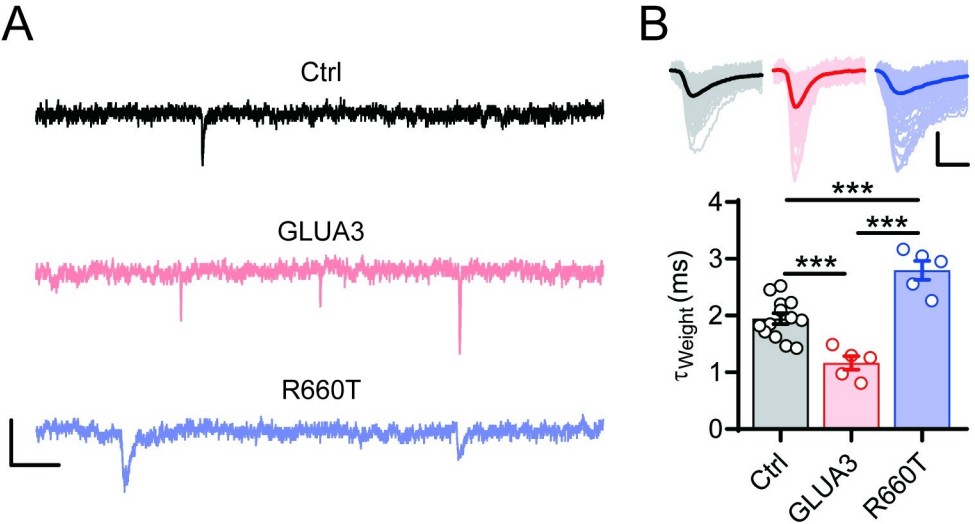

**Fig 5. GLUA3_R660T slows mEPSC decay in CGNs.** (A) Sample mEPSC recordings of untransfected (black), GLUA3-transfected (pink) and GLUA3_R660T-transfected (purple) CGNs. Scale bars, 10 pA and 50 ms. (B) The upper traces are superimposed individual mEPSC traces (light) recorded from signle neurons and the average traces (dark). Scale bars, 10 pA and 2 ms. The lower bar graph presents weighted decay tau of mEPSCs. Circles are the values calculated from average traces of individual neurons. Ctrl, 1.9 ± 0.1 ms, n = 13. GLUA3, 1.2 ± 0.1 ms, n = 5. R660T, 2.8 ± 0.2 ms, n = 5. ***p < 0.001. One-way ANOVA.

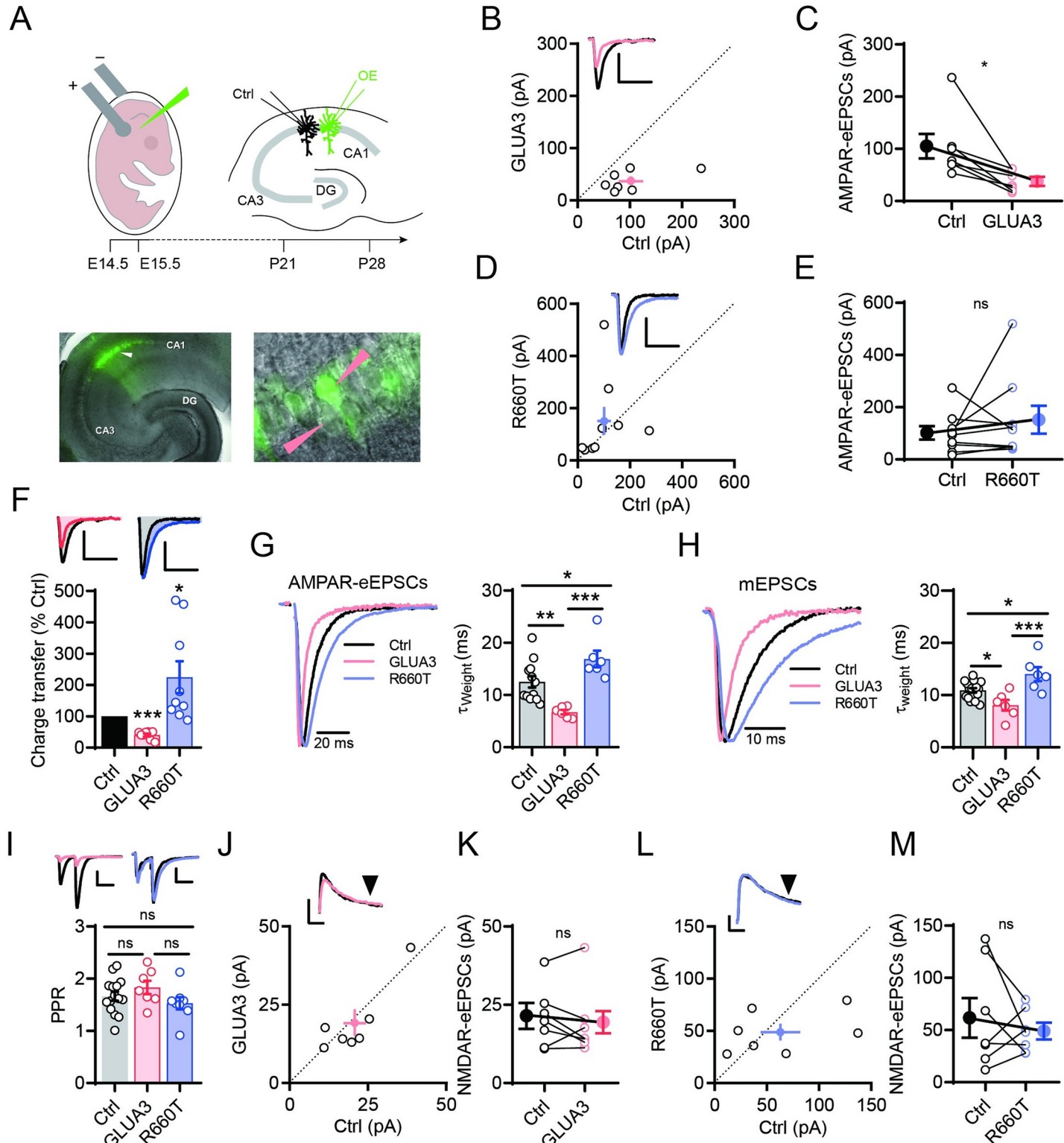

**Fig 6. Overexpression of GLUA3 and R660T regulate AMPAR-EPSC decay in hippocampal CA1 pyramidal neurons.** (A) Up panel, schematic of IUE and dual whole-cell recordings in hippocampal CA1 neurons. Lower left panel, photographs of GLUA3-IRES-GFP transfected CA1 pyramidal neurons on hippocampal acute slice. Lower right panel shows the magnified image around the white arrowhead at the left panel. Pink arrowheads point to a transfected neuron (Green) and a control neuron (no color) that were selected for dual recording. (B) Circles present dual recordings of evoked AMPAR-eEPSCs of GLUA3-transfected (GLUA3) and control (Ctrl) neurons. Filled doc with bars is mean ± SEM of the recordings. Inserted sample traces are from Ctrl (black) recording paired with GLUA3 (pink). Scale bars, 30

pA and 50 ms. (C) Line graph of AMPAR-eEPSCs in (B). Lines link the GLUA3 and Ctrl neurons of individual pairs. Filled docs with bars presented mean ± SEM, Ctrl, 101.9 ± 23.5 pA, GLUA3, 37.6 ± 7.3 pA, n = 7 pairs, *p = 0.019. Paired t-test. (D-E) AMPAR-eEPSCs of GLUA3_R660T overexpressing (R660T) and Ctrl neurons. Inserted sample traces are from Ctrl (black) recording paired with R660T (purple). Scale bars, 30 pA and 50 ms. Filled docs with bars in (E) presented mean ± SEM, Ctrl, 101.1 ± 26.1 pA, R660T, 150.2 ± 52.4 pA. Ctrl vs R660T, n = 9 pairs, ns, p = 0.386. Paired t-test. (F) Bar graph shows charge transfer mediated by AMPAR-eEPSC of transfected neurons relative to relevant Ctrl. Scale bars of inserted sample traces, 30 pA and 50 ms. GLUA3, 40.3 ± 5.1% of Ctrl, n = 7 pairs, ***p < 0.001; R660T, 225.1 ± 50.8% of Ctrl, n = 9 pairs, *p = 0.039. Paired t-test. (G) The decay of AMPAR-eEPSCs. Left, sample traces scaled to peak amplitudes. Right, bar graph shows the weighted tau of Ctrl, GLUA3 and R660T. Ctrl, 12.5 ± 1.1 ms n = 12, GLUA3, 6.7 ± 0.4 ms, n = 6, R660T, 16.9 ± 1.6 ms, n = 6. Ctrl vs GLUA3, **p = 0.007. Ctrl vs R660T, *p = 0.046. GLUA3 vs R660T, ***p < 0.001. One-way ANOVA. (H) The decay of mEPSCs. Left, sample traces of averaged mEPSCs of individual neurons scaled to the peak amplitudes. Right, bar graph shows the weight tau of mEPSC (mean ± SEM). Ctrl, 10.9 ± 0.4 ms, n = 16, GLUA3, 8.0 ± 1.0 ms, n = 6, R660T, 14.0 ± 1.3 ms, n = 6. Ctrl vs GLUA3, *P = 0.038. Ctrl vs R660T, *p = 0.020. GLUA3 vs R660T, ***p < 0.001. One-way ANOVA. (I) Bar graph of paired-pulse ratio. Data are in mean ± SEM. Ctrl, 1.66 ± 0.09, n = 15; GLUA3, 1.83 ± 0.13, n = 7; R660T, 1.53 ± 0.12, n = 8. Ctrl vs GLUA3, ns, p = 0.520; Ctrl vs R660T, ns, p = 0.641; GLUA3 vs R660T, ns, p = 0.208. One-way ANOVA. Inserted sample traces are from Ctrl (black) recording paired with GLUA3. Scale bars, 30 pA and 50 ms. (J) Circles present NMDAR-eEPSCs in dual recordings of GLUA3 and Ctrl neurons measured at 150ms after stimulation at +40mV. Scale bars of inserted sample traces, 30 pA and 50 ms. (K) Line graph of NMDAR-eEPSCs in GLUA3 and control dual recordings. Lines link the GLUA3 and Ctrl neurons of individual recordings. Ctrl, 20.8 ± 3.6 pA, GLUA3, 19.2 ± 4.1 pA. Ctrl vs GLUA3, n = 7, ns, p = 0.470, paired t-test. (L,M) NMDAR-eEPSCs of GLUA3_R660T-overexpressing (R660T) and Ctrl neurons. Scale bars, 30 pA and 50 ms. Filled docs with bars in (M) present mean ± SEM, Ctrl, 62.9 ± 19.1 pA, R660T, 48.7 ± 7.7 pA. Ctrl vs R660T, n = 7. Ctrl vs R660T, ns, p = 0.446, paired t-test.

providing tailored molecular diagnosis enabling precision medicine in approximately one quarter of patients, illustrating the enormous utility of genetic testing for therapeutic decision-making [41,42]. A major goal of genetic studies is the identification of novel drug targets and tailored therapies based on the etiology of disease. The discovery of specific genetic variants has also helped us to repurpose drugs with specific actions which may have been used in entirely unrelated conditions.

In the case of *GLUA3* loss-of-function variants there are no FDA or MDA approved treatment options, however in the case of GoF variants, one drug is available: Perampanel (PER) which is an orally active, selective, non-competitive alpha-amino-3-hydroxy-5-methyl-4-isoxazolepropionic acid receptor agonist [43]. Although there are several publications describing the safety and efficacy of PER for individuals with epilepsy [43–45], there are no similar data available in individuals with *GRIA3* GoF variants. An empirical antiepileptic treatment in monogenic disorders including *GRIA3*-deficiency is often found ineffective, may cause unwanted side effects and thus ultimately result in a diminished quality of life. This is also demonstrated by our case which highlights the urgency for precision treatment options. The GoF property of GLUA3_R660T suggests it enhances and expands excitatory glutamate signaling in brain. Functional analysis on the mutants would lead to some clues on the therapy strategy. Our study suggests that the slow decay kinetics is likely the most relevant characteristic of the mutant channel. Since auxiliary subunit TARPs and cornichons are likely the most important factors in regulating gating kinetics, therapies targeting those auxiliary subunits may represent a promising field.

## Materials and methods

### Ethics statement

The study was conducted in agreement with the Declaration of Helsinki and approved by the Ethics Committee of the Strasbourg University Hospital, approval number CE-2021-72. Formal consent for genetic testing and publication was obtained from the parents. The detailed information about seisure semiology, neurologic examination (EEG and MRI) and treatment outcomes were collected following interview with parents and by reviewing the proband medical files.

### Whole genome sequencing

Whole genome sequencing was performed on extracted DNA using sequencing-by-synthesis (SBS) next generation sequencing (NGS) according to the test definition of the TruGenome Undiagnosed Disease at the time testing was pursued. The data were aligned and reported

according to build 37.1 of the Human Reference Genome (http://www.ncbi.nlm.nih.gov/projects/genome/assembly/grc/human/). The genome was sequenced to an average of ≥30 fold coverage. Over 99% of the genome was covered at 10-fold coverage or more and at least 97% of the genome was callable (passes all quality filters). Based on the quality filters and through the analysis of an extended, multi-generation family set (Platinum Genomes, [46]), for SNVs, sensitivity is 98.9% and the analytical Positive Predictive Value (PPV), i.e. true positive/[true positive + false positive] is 99.9%. Insertions up to 31 bases and deletions up to 27 bases have a sensitivity and analytical PPV of approximately 80–85%, determined through Platinum Genomes. Copy number events greater than 10 kb may be detected; sensitivity for events greater than 20 kb is approximately 85%. For SNVs and small insertion and deletion events, interpretation is limited to variant positions that overlap an exon and its 15 base pair flanking sequence. For CNVs, interpretation is limited to events that either overlap an exon or have a boundary that lies within 1 kb upstream or downstream of an exon. Mitochondrial SNVs detected at an allele fraction greater than or equal to 3% were interpreted for pathogenicity. However, percentage of heteroplasmy is not reported. Mitochondrial CNVs and small insertion and deletions are not reported.

To identify candidate variants of potential clinical interest, variants were filtered and considered based on multiple factors including population allele frequency, variant consequence, evolutionary conservation, occurrence in a gene with a well-established gene-disease relationship, occurrence in a gene whose disease association overlaps with the patient's reported phenotype, and inheritance mode, as appropriate. Clinical interpretation was performed on variants and CNVs of interest in accordance with the American College of Medical Genetics and Genomics guidelines.

The TruGenome Undiagnosed Disease test was developed and its performance characteristics determined by Illumina Clinical Services Laboratory (CLIA# 05D1092911/CAP# 7217613). It has not been cleared or approved by the U.S. Food and Drug Administration. Pursuant to the requirements of CLIA '88, this laboratory test has established and verified the test's accuracy and precision.

## cDNA constructs

cDNAs encoding human GLUA3 was subcloned into the NheI and XhoI restriction sites of the vector pCAGGS-IRES-EGFP. Human GLUA2, TARP γ-2 and CNIH2 were subcloned into the vector pCAGGS-IRES-mCherry. Coexpression of AMPARs and the auxiliary subunits were identified by the fluorescence of EGFP and mCherry. GLUA3_R660T was made by overlapping PCR and confirmed by Sanger sequencing.

## HEK cells

HEK293T cells were cultured in a 37°C incubator supplied with 5% $CO_2$. Transfection was performed in 35-mm dishes using lipofectomine2000 reagents (Invitrogen). When coexpression was carried out, the ratio of GLUA3 to GLUA2 cDNA was 1:1. The ratio between GLUAs and CNIH2 or TARP γ-2 was 1:1. NBQX (100 μM) was included in culture media to block AMPAR-induced cytotoxicity. Cells were dissociated with 0.05% trypsin and plated on coverslips pretreated with poly-D-lysine 24 h post transfection. Recording was performed 4 h after plating.

## Animals

ICR mice were obtained from Gempharmatech Inc. (Nanjing). The mice were maintained in the core animal facility of the Model Animal Research Center (MARC) at Nanjing University

with the room temperature and the light-cycle automatically controlled (25±1˚C; 12 hrs for light and 12 hrs for dark). Mice had free access to food and water. Mouse breeding and experiments were conducted under an IACUC approved protocol. All the experiments were performed in accordance with the Guide for the Care and Use of Laboratory Animals of Nanjing University.

### *In Utero* electroporation

*In utero* electroporation was performed as recently described [47]. E14.5–15.5 pregnant mice were anesthetized with a mixture of ketamine (100 mg/kg body weight) and xylazine (5 mg/kg body weight) via intraperitoneal injection. The mice were then subjected to a surgical procedure to expose the uterus. Each embryo was injected with about 2 μL of plasmid DNA mixed with Fast-Green into the left lateral ventricle via a beveled glass micropipette. The embryos were then electroporated with five 42-V pulses of 50 ms, delivered at 1 Hz, using platinum tweezertrodes in a square-wave pulse generator (BTX, Harvard Apparatus). Following electroporation, the embryos were placed back into the abdominal cavity, and the muscle and skin were sutured. Mice were injected i.p. with carprofen (5mg/kg body weight) and monitored until fully awake.

### Cerebellar granule neuron culture

Primary cultures of granule neurons were prepared from p6-8 mouse cerebella and plated at a density of 2.5 X $10^5$ cells/cm$^2$ on coverslips precoated with poly-D-lysine in 24-well dishes. Cultures were maintained in a BME-based medium (Invitrogen) containing 25 mM KCl, 10% FBS, 2 mM L-Glutamine, 2% B27, and 60 μg/ml gentamicin for 2 days. At DIV 3, the culture medium was replaced with a MEM-based medium (Invitrogen) containing 5 mM KCl, supplemented with 5 mg/ml D-glucose, 2 mM L-glutamine, 1% insulin-transferrin-sodium selenite (Sigma), 20 μg/ml gentamicin, and 4 μM cytosine β-D-arabinofuranoside hydrochloride.

### Electrophysiology

Outside-out patches were excised from HEK 293T with 3–5 MΩ borosilicate glass pipettes filled with the following internal solution (in mM): KF 135, KOH 33, MgCl2·6H2O 2, CaCl2 1, EGTA 11, HEPES 10, with pH adjusted to 7.2. Glutamate (10 mM) were dissolved in extracellular solution (in mM): NaCl 140, KCl 2.5, CaCl2·2H2O 2, MgCl2·6H2O 1, HEPES 10mM, Glucose 5. Glutamate pulses of 1 or 500 ms were applied to patches every 5 s using a θ-glass pipettes mounted on a piezoelectric bimorph driven by gravity. Glutamate-induced currents were recorded with membrane potential held at −70 mV. Data were collected with Axon Digidata 1440 data acquisition system and MultiClamp 700B amplifiers (Axon Instruments), with sampling frequency of 100 kHz with filtration over 2 kHz for fast application.

Acute hippocampal slices were prepared at P21-28. Hippocampi were isolated rapidly and placed into 95% O2 / 5% CO2 saturated ice-cold sucrose-based cutting solution (in mM, KCl 2.5, NaH$_2$PO$_4$ 1.25, NaHCO$_3$ 25, D-glucose 10, sucrose 210, Na-ascorbate 1.3, CaCl$_2$ 0.5 and MgSO$_4$ 7) for 350 μm coronal slicing with vibratome (Leica, VT1000S). Slices were then incubated in 34˚C artificial cerebrospinal fluid (ACSF, in mM, NaCl 119, KCl 2.5, NaHCO$_3$ 26.2, NaH$_2$PO$_4$ 1, D-glucose 11, CaCl$_2$-2H$_2$O 2.5 MgSO$_4$ 1.3 glucose 11, saturated with 95% O2/5% CO2) for 20 min and in room temperature ACSF for 60 min, followed by subsequently electrophysiological recording. The internal solution for hippocampal slice whole-cell recording contained (in mM): CsMeSO$_4$ 135.0, NaCl 8.0 HEPES 10.0, EGTA 0.3, QX-314 5.0, Mg-ATP 4.0, Na3-GTP 0.3 and Spermine-Cl$_4$ 0.1, with PH adjusted to 7.3–7.4 with CsOH, with osmotic pressure adjust to 290–295 mOsm. Recording electrodes were 3–6 MΩ borosilicate

glass pipettes. Concentric bipolar stimulation electrode was placed at the Schaffer collateral. For EPSC recording, $GABA_A$ receptor antagonist (100 μM picrotoxin and 10 μM bicucullin) was consistently perfused. AMPAR-eEPSCs and PPR were recording at -70 mV holding, while NMDAR-eEPSCs were obtained 150 ms after stimulation at +40 mV holding. Miniature EPSC was recorded in gap-free mode at -70 mV, and with extra voltage-gated sodium channel antagonist tetrodotoxin (TTX, 0.5 μM).

The mEPSCs of primary cerebellum granule cells were recorded at DIV9-10. Cells were maintained with the extracellular solution containing (in mM, NaCl 145, KCl 5, CaCl2 1, MgCl2 1, D-glucose 5, sucrose 25, HEPES 5, picrotoxin 0.1, D-APV 0.1, tetrodotoxin 0.001, with PH adjusted to 7.3 with NaOH). Whole-cell recording were made with 7–10 MΩ borosilicate glass pipettes filled with the pipette solution containing (in mM, CsCl 140, MgCl 2, EGTA 5, HEPES 10, $Na_3$-GTP 0.3, $Na_2$-ATP 4 and Spermine-$Cl_4$ 0.1, with PH adjusted to 7.3 with CsOH). Miniature EPSC was recorded in gap-free mode at -70 mV while 200 mM sucrose was dissolved in extracellular solution to facilitate miniature events. Data from hippocampal slices and culture CGNs were collected with Axon Digidata 1550 data acquisition system and Multi-Clamp 700B amplifiers (Axon Instruments), with sampling frequency of 10 kHz with filtration over 2 kHz.

## Non-stationary fluctuation analysis

Non-stationary fluctuation analysis was used to analyze AMPAR channel properties. Responses to 1 ms pulse of glutamate (10 mM) were recorded with the frequency of 0.2 Hz for at least 100 traces. The average variance ($\sigma^2$) was calculated from all the traces and fitted with $\sigma^2 = iI - I^2/N + \sigma_0^2$ where $i$ is single channel current, $I$ is the mean current, $N$ is the number of available channels in the patch and $\sigma_0^2$ is the variance of background noise. The probability of opening of the receptor channels in patch at any given point in time is determined by $P_{open} = I/iN$. The single channel conductance ($\gamma$) was calculated with $\gamma = i/(Vh - V_r)$, where $V_h$ is the holding potential of -70 mV, $V_r$ is the reverse potential for AMPARs assumed to be 0 mV.

## Supporting information

**S1 Fig. The deactivation and desensitization kinetics of GLUA3 and GLUA2/A3 and variant in the presence of CNIH2.** (A). Deactivation of GLUA3 and GLUA3_R660T in the presence of CNIH2. GLUA3 and GLUA3_R660T were the same as in Fig 1. GLUA3/CNIH2, 8.6 ± 1.4 ms, n = 9; R660T/CNIH2, 15.1 ± 1.9 ms, n = 9; *p = 0.0136. (B) Up, the sample traces for desensitization of GLUA2/A3 and GLUA2/A3_R660T in the presence of CNIH2. Low left, statistics of weighted $\tau_{des}$. GLUA3/CNIH2, 49.0 ± 6.0 ms, n = 8; R660T/CNIH2, no clear desensitization and $\tau_{des}$ value cannot be calculated, n = 9. Low right, statistics of steady state currents. GLUA3/CNIH2, 15.9 ± 3.0%, n = 7; R660T/CNIH2, 79.1 ± 3.2%, n = 7; ***p < 0.001. (C) Deactivation of GLUA2/A3 and GLUA2/A3_R660T in the presence of CNIH2. GLUA2/A3 and GLUA2/A3_R660T were the same as in Fig 2. GLUA2/A3/CNIH2, 5.0 ± 1.0 ms, n = 9; GLUA2/A3_R660T/CNIH2, 16.7 ± 1.7 ms, n = 9; ***p < 0.001. (D) Up, the sample traces for desensitization of GLUA2/A3/CNIH2 and GLUA2/A3_R660T/CNIH2. Low left, statistics of weighted $\tau_{des}$. GLUA2/A3/CNIH2, 33.0 ± 2.0 ms, n = 9; GLUA2/A3_R660T/CNIH2, 52.4 ± 6.1 ms, n = 9; **p = 0.008. Low right, statistics of steady state currents. GLUA2/A3/CNIH2, 17.5 ± 2.9%, n = 9; GLUA2/A3_R660T/CNIH2, 66.8 ± 4.3%, n = 9; ***p < 0.001. Data are presented as mean ± SEM. Unpaired t-test was used for data analysis.
(TIF)

## Acknowledgments

Clinical whole genome sequencing (cWGS) was provided through the Illumina iHope Program, a philanthropic initiative supported by Illumina, Inc. The iHope Program donates cWGS testing for use in patients referred for genetic diagnosis by their healthcare providers, but who lack access to this testing.

## Author Contributions

**Conceptualization:** Diane Masser-Frye, Marilyn Jones, Leslie Patron Romero, Berardo Rinaldi, Wenhui Laura Li, Benedicte Gerard, Erin Thorpe, Allan Bayat, Yun Stone Shi.

**Data curation:** Jia-Hui Sun, Jiang Chen, Fernando Eduardo Ayala Valenzuela, Carolyn Brown, Qing-Qing Li, Dan Wu.

**Formal analysis:** Jia-Hui Sun, Jiang Chen, Carolyn Brown.

**Funding acquisition:** Yun Stone Shi.

**Investigation:** Jia-Hui Sun, Jiang Chen, Fernando Eduardo Ayala Valenzuela, Carolyn Brown.

**Methodology:** Carolyn Brown, Leslie Patron Romero, Berardo Rinaldi, Erin Thorpe.

**Project administration:** Erin Thorpe, Yun Stone Shi.

**Resources:** Fernando Eduardo Ayala Valenzuela, Wenhui Laura Li.

**Supervision:** Allan Bayat, Yun Stone Shi.

**Writing – original draft:** Erin Thorpe, Allan Bayat, Yun Stone Shi.

**Writing – review & editing:** Allan Bayat, Yun Stone Shi.

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
