## [Decision Letter · Decision Letter 0]

14 Apr 2021

Dear Dr Shi,

Thank you very much for submitting your Research Article entitled 'X-linked neonatal-onset epileptic encephalopathy associated with a gain-of-function variant p.R660T in GRIA3' to PLOS Genetics.

The manuscript was fully evaluated at the editorial level and by independent peer reviewers. The reviewers appreciated the attention to an important topic but identified some concerns that we ask you address in a revised manuscript

We therefore ask you to modify the manuscript according to the review recommendations. Your revisions should address the specific points made by each reviewer.

[LINK]

Yours sincerely,

Wei Zhang

Guest Editor

PLOS Genetics

Hua Tang

Section Editor: Natural Variation

PLOS Genetics

Reviewer's Responses to Questions

**Comments to the Authors:**

Reviewer #1: In the Manuscript entitled "X-linked neonatal-onset epileptic encephalopathy associated with a gain-of-function variant p.R660T in GRIA3", Sun J. et al. identified a pathogenic missense variant in GRIA3 gene in a female patient with severe epilepsy and global developmental delay. Through electrophysiological analysis the authors found this mutant makes strengthened glutamate receptors, which indicates that the variant causes epileptic encephalopathy and global developmental delay by enhancing glutamate signaling in brain. This is a very interesting story and set a good example for the combination for clinical and basic research. Please address the following questions before being considered for publication.

1. Other than the little girl mentioned in this study, how many more patients are found carrying this A3 mutation?

2. After whole genome sequencing, how many site mutation(s) or SNPs are discovered in the little girl? If more mutations are found, what roles do those mutations play in epilepsy?

3. In figures 5 and 6, what roles does endogenous GluA3 play?

Reviewer #2: In this study the authors identified a female individual exhibiting epileptic encephalopathy and harboring a missense mutation in GRIA3 gene resulting in a GluA3 R660T protein variant. Expression of the R660T GluA3 mutant alone in HEK cells resulted in a marked slowing of GluA3 homomer deactivation and desensitization kinetics and produced a large increase in steady state current of the channel. This mutation produced similar effects in GluA2/A3 heteromeric channels in HEK cells. The effects of this mutation were also present with GluA2 homomers and GluA2/A3 heteromers associated with y-2 and CNIH2 auxiliary subunits. Additionally, overexpression of GluA3 R660T in cerebellar granule neurons produced slower mEPSC decay kinetics when compared to overexpression of wild-type GluA3. GluA3 R660T overexpression in CA1 pyramidal neurons was also shown to increase AMPAR-eEPSC charge transfer and slow AMPAR-eEPSC and mEPSC decay kinetics compared to wild-type GluA3 overexpression.

Overall this is a very interesting and carefully executed study that clearly describes the nature of the glutamatergic synapse perturbation that likely contributes to this individual’s epileptic encephalopathy. While a molecular replacement approach where GluA3 is removed and replaced with GluA3 R660T might be preferable to the overexpression expression experiments performed, I do not regard this as a major limitation of the study. From the work the authors’ have already done it is very clear what this mutation is doing to AMPAR function. I would have no problem with the manuscript being published in its current form.

Minor issues:

1) Were mutations found in any other genes associated with epilepsy in this individual?

2) In the first paragraph of the introduction “Though GLUA3 alone can form Ca2+ permeable homomeric receptors, in the brain it mainly forms heteromeric receptors with GLUA2” needs a reference.

**Have all data underlying the figures and results presented in the manuscript been provided?**

Reviewer #1: Yes

Reviewer #2: Yes

PLOS authors have the option to publish the peer review history of their article (what does this mean?). If published, this will include your full peer review and any attached files.

Reviewer #1: No

Reviewer #2: No

---

## [Decision Letter · Decision Letter 1]

18 May 2021

Dear Dr Shi,

We are pleased to inform you that your manuscript entitled "X-linked neonatal-onset epileptic encephalopathy associated with a gain-of-function variant p.R660T in GRIA3" has been editorially accepted for publication in PLOS Genetics. Congratulations!

Yours sincerely,

wei zhang

Guest Editor

PLOS Genetics

Hua Tang

Section Editor: Natural Variation

PLOS Genetics

Comments from the reviewers (if applicable):

Reviewer's Responses to Questions

**Comments to the Authors:**

Reviewer #1: The authors have addressed all my questions, I have no further comments.

Reviewer #2: The authors have addressed all of my concerns. I am happy to recommend the manuscript for publication in its current form.

**Have all data underlying the figures and results presented in the manuscript been provided?**

Reviewer #1: Yes

Reviewer #2: Yes

PLOS authors have the option to publish the peer review history of their article (what does this mean?). If published, this will include your full peer review and any attached files.

Reviewer #1: No

Reviewer #2: No

**Data Deposition**

http://datadryad.org/submit?journalID=pgenetics&manu=PGENETICS-D-21-00291R1

**Press Queries**

---

## [Editor Report · Acceptance letter]

18 Jun 2021

PGENETICS-D-21-00291R1 

X-linked neonatal-onset epileptic encephalopathy associated with a gain-of-function variant p.R660T in *GRIA3*  

Dear Dr Shi, 

We are pleased to inform you that your manuscript entitled "X-linked neonatal-onset epileptic encephalopathy associated with a gain-of-function variant p.R660T in *GRIA3* " has been formally accepted for publication in PLOS Genetics! Your manuscript is now with our production department and you will be notified of the publication date in due course.

With kind regards,

Agota Szep

PLOS Genetics

On behalf of:
